# Orbit-Augmented Geometric Consistency for Lunar Neural Elevation Model

Minseok Song     Suwan Lee     Seokju Lee[†]

Korea Institute of Energy Technology (KENTECH)

{shurup11, bigtiger0729, slee}@kentech.ac.kr

## Abstract

*Reconstructing high-resolution lunar digital elevation models (DEMs) from sparse orbital imagery remains challenging due to limited viewpoint diversity and strong illumination variations. We present an extension of the Lunar Neural Elevation Model (LNEM) that incorporates orbit-aware data augmentation and multi-orbit geometric consistency, improving reconstruction robustness under sparse-view conditions. Our approach synthesizes additional training views via orbit-conditioned sampling while preserving pushbroom imaging geometry. To enforce cross-orbit consistency, we introduce a geometric constraint based on bidirectional reprojection between orbit-derived views, enabling stable multi-view alignment despite limited parallax. The proposed framework integrates orbit augmentation and geometric consistency into a unified training pipeline without requiring dense image coverage. Experiments on LROC NAC data show that the proposed constraint improves geometric consistency and reduces bias-corrected LOLA RMSE relative to the LNEM baseline without geometric consistency. We further report NAC DTM and SLDEM as contextual production DEM references, rather than as direct baselines to be surpassed. These results demonstrate the effectiveness of orbit-aware augmentation and multi-orbit geometric constraints for sparse-view pushbroom lunar reconstruction.*

## 1. Introduction

High-resolution lunar digital elevation models (DEMs) provide essential information for landing site analysis, rover path planning, and localized geological interpretation [7]. Broadly, lunar topographic reconstruction has been pursued through three primary modalities. Orbital laser altimetry, exemplified by LOLA [19], provides highly accurate but spatially sparse elevation. Stereo photogrammetry, including ASP [2] and LROC NAC stereo reconstruction [7],

---

[†]Corresponding author.

extracts elevation from multi-view parallax but struggles in textureless regions. Shape-from-shading methods [10] infer topography from photometric cues but are sensitive to albedo variations. Although these approaches provide complementary strengths, robust high-resolution lunar topographic reconstruction remains difficult in practice, especially when observations are limited and illumination conditions vary substantially across acquisitions.

Among these challenges, sparse-view reconstruction from orbital pushbroom imagery is particularly difficult. Current high-resolution topographic modeling relies heavily on pushbroom sensors such as the LROC Narrow Angle Camera (NAC) [18] and KPLO Lunar Terrain Imager (LUTI) [9]. However, only a limited number of orbital passes are often available over a target region, leading to under-constrained reconstruction with insufficient parallax and limited viewpoint diversity. Moreover, pushbroom imagery differs fundamentally from standard pinhole cameras. In linear pushbroom cameras [5], each scanline has its own projection center and sensor attitude. This line-wise geometry is explicitly handled in pushbroom stereo pipelines [3]. This dynamic acquisition geometry, combined with varying illumination across orbits, makes cross-view geometric consistency difficult to enforce.

Recently, the Lunar Neural Elevation Model (LNEM) [12] introduced a volumetric neural reconstruction baseline tailored for lunar observations by incorporating Rigorous Sensor Models (RSMs) and line-wise orbital geometry. While LNEM provides a more physically grounded alternative to conventional RPC-based satellite NeRF formulations such as Sat-NeRF [14] and EO-NeRF [15], sparse-view conditions still weaken cross-view geometric constraints. As a result, reconstruction tends to rely more heavily on per-ray photometric consistency, which can increase geometric variance in the reconstructed 3D point cloud.

To mitigate this geometric instability, we propose multi-orbit geometric consistency for sparse-view pushbroom reconstruction, together with orbit-aware viewpoint augmentation. We synthesize additional views by applying Gaussian-perturbed orbit-level transformations to each or-

bital trajectory, while performing scanline-level attitude repointing to preserve the original region-of-interest track. Using both observed and orbit-augmented views, we formulate a bidirectional reprojection geometric consistency constraint under line-wise pushbroom geometry. The constraint establishes cross-view correspondences and encourages the corresponding 3D surface points to agree across views, improving reconstruction stability under sparse-view conditions without requiring additional real observations.

The main contributions of this work are as follows:

- **Gaussian orbit augmentation.** We propose a Gaussian orbit augmentation strategy that synthesizes augmented views for sparse-view pushbroom lunar reconstruction while preserving line-wise acquisition geometry.
- **Geometric consistency constraint.** We introduce a geometric consistency constraint tailored to line-wise pushbroom imaging. The constraint uses bidirectional reprojection between orbit-derived views to enforce surface-level cross-view geometric consistency under sparse-view conditions.
- **Empirical topographic validation.** We show that the proposed geometric consistency constraint reduces multi-view 3D coordinate standard deviation, indicating improved cross-view geometric stability under sparse pushbroom observations. The same constraint also reduces bias-corrected LOLA RMSE relative to the ablated LNEM variant without geometric consistency.

## 2. Related Work

### 2.1. Lunar DEM Reconstruction and Pushbroom Geometry

Lunar DEM reconstruction has traditionally relied on complementary modalities including orbital laser altimetry, stereo photogrammetry, and shape-from-shading. Orbital laser altimetry, represented by LOLA [19], provides highly reliable elevation measurements with excellent vertical accuracy and therefore serves as a key geometric reference for lunar topography. However, its sparse spatial sampling limits its ability to recover high-resolution local terrain as a standalone source. In contrast, optical stereo reconstruction from LROC NAC observations [7] enables high-resolution DEM generation over selected local regions and remains a strong practical reference for lunar photogrammetric reconstruction. SLDEM [1] combines LOLA and SELENE Terrain Camera observations to achieve broader regional coverage, illustrating the long-standing trade-off between spatial coverage and local resolution in lunar DEM products. In addition, the Ames Stereo Pipeline (ASP) [2] provides a general photogrammetric framework for DEM generation from planetary and satellite imagery and has been broadly used in lunar topographic processing. Shape-from-shading approaches [10] have also been explored for lunar terrain re-

covery from single-image photometric cues, particularly in challenging regions such as the lunar south pole. However, these methods remain sensitive to illumination uncertainty, reflectance assumptions, and albedo variation.

For high-resolution lunar observations, reconstruction increasingly depends on orbital pushbroom sensors such as LROC NAC [18] and KPLO LUTI [9]. Unlike standard pinhole cameras, pushbroom sensors do not admit a single global projection center. In linear pushbroom cameras [5], both the projection center and sensor attitude vary across scanlines. Pushbroom stereo pipelines [3] explicitly handle this line-wise geometry. More broadly, pushbroom imaging belongs to the family of multiperspective imaging systems, whose geometry departs fundamentally from conventional perspective assumptions [24]. As a result, pushbroom imagery does not admit a standard global epipolar geometry, and correspondence search, reprojection, and cross-view alignment must be handled in a line-wise rather than image-wise manner. Under sparse multi-orbit observations, these geometric properties make accurate reconstruction substantially more difficult and motivate methods that explicitly model pushbroom-specific acquisition geometry.

### 2.2. Neural Rendering for Satellite and Lunar Imagery

NeRF [16] introduced a continuous volumetric scene representation for high-quality novel view synthesis and 3D reconstruction from images. Since then, neural rendering has been extended to a range of remote sensing and satellite imaging settings. In the satellite domain, representative approaches such as Sat-NeRF [14] and EO-NeRF [15] incorporate RPC camera models together with illumination-aware modeling to reconstruct terrain from multi-view observations acquired under varying temporal and lighting conditions. These studies demonstrate the potential of neural volumetric reconstruction for overhead imagery. However, they adopt RPC-based imaging formulations that do not explicitly capture the line-wise acquisition geometry described in classical pushbroom camera models [5] and pushbroom stereo formulations [3]. As a result, their representations do not fully reflect the scanline-dependent imaging geometry inherent in lunar pushbroom observations.

Neural rendering for lunar surface reconstruction has also begun to emerge. Recent work has explored neural radiance methods for lunar terrain modeling under sparse observations and discussed both their potential and their limitations in planetary settings [20]. LNEM [12] advances this direction by introducing a volumetric reconstruction framework tailored to lunar pushbroom imagery through the integration of Rigorous Sensor Models (RSMs) and line-wise orbital geometry. Compared with RPC-based satellite formulations, LNEM provides a more physically grounded treatment of lunar orbital imaging and establishes a strong

baseline for neural DEM reconstruction from real pushbroom observations. However, physically grounded sensor modeling alone does not fully resolve sparse-view ambiguity. When angular redundancy is limited, the learned volumetric representation can still admit view-inconsistent surface hypotheses, leading to increased 3D variance and weaker cross-view geometric agreement. Our work builds directly on LNEM and targets this remaining limitation by strengthening geometric supervision under sparse-view pushbroom observations.

### 2.3. Sparse-View Regularization and Geometric Supervision for Neural Reconstruction

Sparse-view neural reconstruction is fundamentally challenged by insufficient parallax and limited viewpoint diversity, which weaken geometric constraints and increase ambiguity in the recovered 3D structure. Prior work has therefore introduced a variety of regularization and auxiliary supervision strategies to stabilize reconstruction from sparse inputs. RegNeRF [17] improves sparse-input reconstruction by regularizing geometry and appearance from unobserved viewpoints. DS-NeRF [4] shows that even sparse depth supervision can significantly improve optimization efficiency and geometric fidelity in view-sparse settings. GeoNeRF [11] incorporates geometry priors to improve sparse-view reconstruction and generalization. FreeNeRF [23] mitigates overfitting in few-shot neural rendering through frequency-space regularization, while SparseNeRF [21] leverages ambiguous depth priors through depth ranking and spatial continuity constraints for few-shot novel view synthesis. Collectively, these methods share a common principle: when photometric supervision alone is insufficient, additional geometric constraints, priors, or regularization are needed to reduce ambiguity in 3D reconstruction.

Our work is conceptually related to this line of research, but differs in the geometric entity being regularized. In pinhole camera settings, sparse-view supervision is typically defined over image-level camera poses, where each view is parameterized by a single rotation matrix $\mathbf{R}$ and translation vector $\mathbf{t}$. Under this formulation, reprojection and viewpoint augmentation can be naturally expressed at the image level. In contrast, pushbroom imagery is governed by time-varying position and orientation functions along the orbital trajectory, denoted by $\mathcal{P}(\tau)$ and $\mathcal{R}(\tau)$, where $\tau$ corresponds to the acquisition time of each scanline, rather than a single global pose per image. This distinction makes the direct application of existing sparse-view regularization methods nontrivial. The central principle of enforcing stronger geometric consistency must therefore be reformulated from image-level pose augmentation and reprojection to line-wise augmentation and reprojection defined along the orbital trajectory. To address this gap, we intro-

duce Gaussian orbit augmentation together with a bidirectional reprojection-based geometric consistency constraint for pushbroom imaging, thereby improving geometric consistency under sparse lunar observations.

## 3. Methodology

We address sparse-view lunar surface reconstruction from a small number of pushbroom orbital images. Unlike standard pinhole camera settings, each pushbroom observation follows the line-wise imaging geometry of linear pushbroom cameras [5]. In this setting, the projection center and sensor attitude vary across scanlines, and this line-wise geometry is explicitly handled in pushbroom stereo pipelines [3]. Under sparse-view conditions, limited parallax and viewpoint diversity weaken cross-view geometric constraints and cause reconstruction to rely more heavily on per-ray photometric consistency. To mitigate this limitation, we introduce Gaussian orbit augmentation together with a geometric consistency constraint for pushbroom imaging. The proposed method consists of four components: Gaussian orbit augmentation, scanline-wise attitude repointing, continuous pushbroom reprojection, and bidirectional multi-orbit geometric supervision.

### 3.1. Gaussian Orbit Augmentation

To compensate for limited viewpoint diversity under sparse-view conditions, we synthesize orbit-augmented views by augmenting each pushbroom orbit as a whole rather than modifying scanlines independently. This design preserves coherent line-wise viewing geometry while generating orbit-augmented views over the same target terrain. As illustrated in Fig. 1, the proposed augmentation produces shifted pushbroom trajectories that still observe the same region, thereby providing additional geometric constraints without requiring extra real observations. Rather than adopting a fully unconstrained 3D rigid transformation, we construct each augmented orbit using a progress-scaled base shift together with anisotropic Gaussian translation noise, while restricting orientation augmentation to a roll offset around the mean forward axis of the orbit. To keep the augmentation tied to scene geometry, translational quantities are parameterized in scene-relative ratio units and converted to metric offsets using a scene scale $s_{\text{scene}}$. Let $k$ denote the training step index, which controls the augmentation magnitude through the progress-dependent scale $\alpha(k)$. The translational augmentation is defined as

$$\Delta\mathbf{t}^{(k)} = s_{\text{scene}}\,\alpha(k)\left(\mathbf{b}_{\text{tr}} + \boldsymbol{\epsilon}_{\text{tr}}\right), \qquad \boldsymbol{\epsilon}_{\text{tr}} \sim \mathcal{N}(\mathbf{0}, \boldsymbol{\Sigma}_{\text{tr}}) \tag{1}$$

where $\mathbf{b}_{\text{tr}} \in \mathbb{R}^3$ denotes a predefined base shift in scene-relative coordinates, $\alpha(k)$ is a progress-dependent scale factor at training step $k$, and $\boldsymbol{\Sigma}_{\text{tr}}$ is an anisotropic covariance matrix for the translation noise. For orientation augmen-

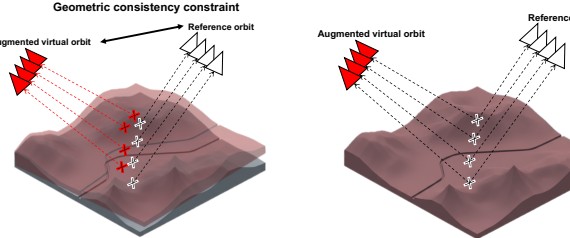

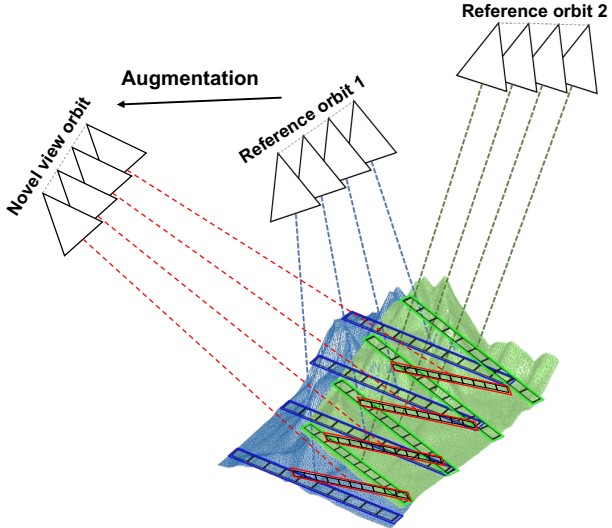

Figure 1. **Illustration of the proposed Gaussian orbit augmentation.** A pushbroom observation is augmented at the orbit level to synthesize orbit-augmented views with shifted viewing geometry over the same target terrain. This orbit-level design preserves coherent line-wise geometry while increasing viewpoint diversity under sparse-view conditions.

tation, we apply only a roll offset around the orbit's mean forward axis $\bar{f}$:

$$\Delta\phi^{(k)} = \alpha(k)\,\epsilon_\phi, \qquad \epsilon_\phi \sim \mathcal{N}\big(0, \sigma_\phi^2\big) \qquad (2)$$

where $\sigma_\phi^2$ controls the magnitude of the roll offset.

This formulation preserves the overall along-track structure of the original pushbroom trajectory while progressively exposing the model to larger viewpoint variations during training. Early in training, the augmentation magnitude remains small so that the model first observes conservative virtual views. As training progresses, the augmentation scale increases, allowing the model to learn from a broader range of geometrically plausible viewing configurations. In implementation, we maintain a small cache of orbit augmentations indexed by orbit slots and reuse them for several training steps before refreshing them, rather than precomputing a fixed augmented set offline.

### 3.2. Scanline-Level Attitude Repointing

Because pushbroom cameras do not admit a single global camera pose [5], orbit-level augmentation alone can introduce geometric drift from the original region of interest. This line-wise property is also central to pushbroom stereo pipelines [3]. To preserve the target ground track, we perform scanline-wise attitude repointing on the augmented orbit. Specifically, each scanline is modeled as a 1D pinhole camera, whose orientation is adjusted such that its center ray is aligned with a 3D ground anchor point obtained from

(a) Without geometric consistency constraint.

(b) With geometric consistency constraint.

Figure 2. **Intuition of the proposed geometric consistency constraint.** Different sparse-view pushbroom observations of the same lunar region may reconstruct slightly inconsistent local geometry when optimized mainly from view-dependent photometric cues. The proposed cross-view constraint encourages these observations to converge to a shared surface geometry. Compared with (a), the reconstruction in (b) exhibits improved geometric agreement across observations.

the center pixel of the corresponding reference scanline. This constraint ensures that the central viewing direction of each scanline remains fixed on the original ground location, while the off-center rays are allowed to vary. As a result, the overall ground footprint of each scanline is modified, providing additional viewpoint diversity, while maintaining alignment with the region of interest and consistency with pushbroom projection geometry.

### 3.3. Geometric Consistency Constraint

Motivated by geometric supervision strategies for sparse-view neural reconstruction, we introduce a geometric consistency constraint tailored to the line-wise geometry of pushbroom imaging. As illustrated in Fig. 2, different sparse-view pushbroom observations of the same region may reconstruct slightly inconsistent local geometry when optimization is driven primarily by view-dependent photometric cues. The proposed constraint uses bidirectional reprojection to establish cross-view correspondences and encourages the corresponding surface points to agree on a shared geometry.

A key component of our implementation is that, for each target view, we explicitly construct a per-line pushbroom camera model from the target rays themselves, rather than relying on a single global target-camera approximation. Given a source depth map, we back-project each valid source sample into a 3D surface point using the source line-wise camera geometry. The resulting point is then projected into a target view through continuous pushbroom reprojection, yielding sub-pixel line and sample coordinates. Instead of sampling a scalar target depth at this location, we construct the corresponding target surface point by interpolating neighboring 3D surface points in the Moon body-

fixed frame. This formulation uses reprojection to establish continuous cross-view correspondences, while the consistency loss is defined as a metric surface-point discrepancy.

We evaluate geometric consistency over bidirectional view pairs. Our formulation follows standard multi-view geometry, where image measurements and depth estimates can be lifted to 3D and related across views through projection and reprojection [6]. Similar 3D consistency principles have been used in depth learning and multi-view stereo by checking agreement of reconstructed 3D structures across views [13, 22].

Since pushbroom images follow line-wise camera geometry, the target correspondence is computed using the per-line camera rays of the target view rather than a single global camera pose [3, 5]. We retain continuous target coordinates and use them for interpolation, following the standard use of continuous-coordinate warping in differentiable image sampling [8].

Let $\mathcal{R}$ denote the set of original reference pushbroom views, and let $n$ denote an orbit-augmented view sampled during training. We use two consistency terms. The first term enforces consistency among the original reference views, anchoring the reconstruction to the geometry of the observed pushbroom orbits. The second term enforces consistency between each reference view and the orbit-augmented view, extending the same geometric constraint to synthesized viewpoints. The two terms are categorized according to the composition of each view pair. Reference-view pairs define the first term, and pairs between reference views and orbit-augmented views define the second term. Each selected pair is evaluated bidirectionally using both source-to-target directions.

For a source-to-target direction $a \rightarrow b$, let $\mathcal{V}_{a \rightarrow b}$ denote the set of valid source samples. For each $\mathbf{u} \in \mathcal{V}_{a \rightarrow b}$, the source surface point is:

$$\mathbf{P}_a(\mathbf{u}) = \mathbf{o}_a(\mathbf{u}) + D_a(\mathbf{u})\mathbf{d}_a(\mathbf{u}), \qquad (3)$$

where $\mathbf{o}_a(\mathbf{u})$, $\mathbf{d}_a(\mathbf{u})$, and $D_a(\mathbf{u})$ denote the ray origin, ray direction, and depth of the source view, respectively. When the source is a reference view, $D_a$ denotes the rendered depth for that view; when the source is an orbit-augmented view, it denotes the rendered augmented-view depth.

The source point is projected into the target pushbroom view using the target line-wise camera model, yielding a continuous target coordinate

$$\tilde{\mathbf{u}}_b = (\tilde{l}_b, \tilde{s}_b). \qquad (4)$$

This coordinate is kept continuous rather than rounded to a discrete pixel. At this location, we construct the corresponding target surface point by bilinearly interpolating neighboring target 3D surface samples in the Moon body-

Table 1. **Bias-corrected elevation error (m) with respect to LOLA measurements across two LROC NAC regions.** $\mathrm{RMSE}_{\mathrm{corr}}$ follows the LNEM evaluation protocol and measures the root mean squared elevation error after removing the global vertical bias between the predicted DEM and the LOLA reference elevations: $\mathrm{bias} = \mathrm{median}(z_{\mathrm{LOLA}} - z_{\mathrm{pred}})$ and $\mathrm{RMSE}_{\mathrm{corr}} = \sqrt{\frac{1}{N}\sum_{i=1}^{N}\left(z_{\mathrm{LOLA}}^{(i)} - z_{\mathrm{pred}}^{(i)} - \mathrm{bias}\right)^2}$. SLDEM and NAC DTM are included as reference DEM products for contextual comparison, while the primary comparison is between our model with the geometric consistency constraint and its ablated counterpart without the geometric consistency constraint (GCC).

| Method | Apollo 15 | Tycho |
|---|---|---|
| **SLDEM** [1] | | |
| Pixel Scale (m/px) | 59.0 | 59.0 |
| $\mathrm{RMSE}_{\mathrm{corr}}$ (m) | 1.993 | 2.717 |
| **NAC DTM** [7] | | |
| Pixel Scale (m/px) | 5.0 | 5.0 |
| $\mathrm{RMSE}_{\mathrm{corr}}$ (m) | 0.986 | 0.860 |
| **Ours (w/ GCC)** | | |
| Pixel Scale (m/px) | 1.3 | 1.3 |
| $\mathrm{RMSE}_{\mathrm{corr}}$ (m) | 1.530 | 0.820 |
| **Ours (w/o GCC)** | | |
| Pixel Scale (m/px) | 1.3 | 1.3 |
| $\mathrm{RMSE}_{\mathrm{corr}}$ (m) | 2.621 | 1.013 |

fixed frame:

$$\mathbf{S}_b(\tilde{\mathbf{u}}_b) = \sum_{\mathbf{v} \in \mathcal{N}_b(\tilde{\mathbf{u}}_b)} \omega_{\mathbf{v}}(\tilde{\mathbf{u}}_b)\,\mathbf{P}_b(\mathbf{v}), \qquad (5)$$

where $\mathcal{N}_b(\tilde{\mathbf{u}}_b)$ denotes the four neighboring target samples around the continuous coordinate $\tilde{\mathbf{u}}_b$, and $\omega_{\mathbf{v}}(\tilde{\mathbf{u}}_b)$ are the bilinear interpolation weights. For an original reference target view, the neighboring target surface samples are constructed from the reference slant-depth map converted to 3D points. For an orbit-augmented target view, the neighboring target surface samples are constructed from rendered augmented-view depths.

The directional surface discrepancy is defined as

$$\ell_{a \rightarrow b} = \frac{1}{|\mathcal{V}_{a \rightarrow b}|} \sum_{\mathbf{u} \in \mathcal{V}_{a \rightarrow b}} \rho\left(\frac{\|\mathbf{P}_a(\mathbf{u}) - \mathbf{S}_b(\tilde{\mathbf{u}}_b)\|_2}{s_{\mathrm{geo}}}\right), \quad (6)$$

where $\rho(\cdot)$ is the Smooth-L1 penalty and $s_{\mathrm{geo}}$ is a metric normalization scale. The valid set $\mathcal{V}_{a \rightarrow b}$ includes only source samples whose continuous reprojection lies inside the target image domain and whose local target surface support is valid.

The bidirectional consistency loss for a view pair $(a, b)$ is then defined as

$$\mathcal{L}_{a \leftrightarrow b} = \frac{1}{2}\left(\ell_{a \rightarrow b} + \ell_{b \rightarrow a}\right). \qquad (7)$$

In practice, a direction with no valid support is excluded from the pair average.

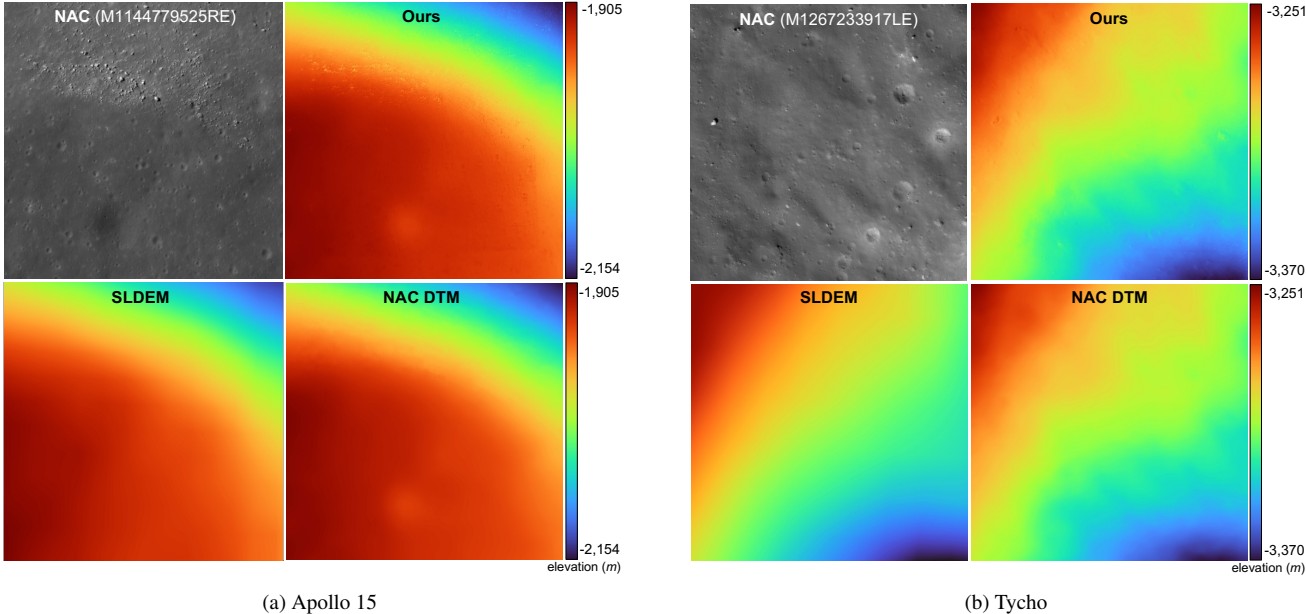

(a) Apollo 15                                                (b) Tycho

Figure 3. **Qualitative comparison with NAC DTM and SLDEM across two lunar regions.** Each subfigure contains the reference NAC image, our neural reconstruction, SLDEM, and NAC DTM. Our reconstruction exhibits smooth local elevation variations and remains consistent with the overall terrain structure of the reference DEM products.

**Reference-view consistency.** The Reference-view consistency term averages bidirectional losses over all pairs of original reference views:

$$\mathcal{L}_{\text{ref}} = \frac{1}{\binom{|\mathcal{R}|}{2}} \sum_{\substack{r_i, r_j \in \mathcal{R} \\ i < j}} \mathcal{L}_{r_i \leftrightarrow r_j}. \tag{8}$$

**Augmented-view consistency.** The augmented-view consistency term averages bidirectional losses between the orbit-augmented view and each original reference view:

$$\mathcal{L}_{\text{aug}} = \frac{1}{|\mathcal{R}|} \sum_{r \in \mathcal{R}} \mathcal{L}_{r \leftrightarrow n}. \tag{9}$$

We combine the two consistency terms as

$$\mathcal{L}_{\text{geometry}} = \frac{w_{\text{ref}}(k)\mathcal{L}_{\text{ref}} + w_{\text{aug}}(k)\mathcal{L}_{\text{aug}}}{w_{\text{ref}}(k) + w_{\text{aug}}(k)}. \tag{10}$$

Here, $w_{\text{ref}}(k)$ and $w_{\text{aug}}(k)$ denote the effective weights of the reference-view and orbit-augmented view consistency terms at training step $k$. A term contributes to the weighted average only when it contains valid projected samples.

### 3.4. Training Objective

Following the photometric reconstruction and depth-supervision framework in LNEM [12], we train the model with image reconstruction, direct depth supervision, and the proposed geometric consistency term:

$$\mathcal{L} = \mathcal{L}_{\text{image}} + \lambda_{\text{depth}}(k)\mathcal{L}_{\text{depth}} + \lambda_{\text{geometry}}(k)\mathcal{L}_{\text{geometry}}. \tag{11}$$

Here, $\mathcal{L}_{\text{image}}$ is the grayscale reconstruction loss and $\mathcal{L}_{\text{depth}}$ is direct slant-depth supervision. The depth weight $\lambda_{\text{depth}}(k)$ follows the same scheduled depth supervision strategy as LNEM, while $\lambda_{\text{geometry}}(k)$ is gradually warmed up after an initial cold-start period so that the geometry term regularizes the field after a coarse surface has been established.

The geometric consistency term $\mathcal{L}_{\text{geometry}}$ is the weighted combination of the reference-view consistency term and the augmented-view consistency term defined in Eq. 10. The reference-view consistency term enforces bidirectional consistency among the original reference views, while the augmented-view consistency term enforces bidirectional consistency between each original reference view and the orbit-augmented view.

For fair ablation, the model without geometric consistency uses the same input observations, depth supervision schedule, and training configuration, with only $\mathcal{L}_{\text{geometry}}$ removed.

## 4. Experiments

### 4.1. Experimental Setup

We evaluate the proposed method on two LROC NAC regions: Apollo 15 and Tycho. Apollo 15 contains diverse

terrain structures such as Hadley Rille and sharp mare–highland transitions, while Tycho contains impact-crater terrain with sharper local relief. For Apollo 15, we use three LROC NAC pushbroom images, namely M183504057LE, M1144779525RE, and M1356476581RE, as input observations. For Tycho, we use three LROC NAC pushbroom images, namely M1267226882LE, M1267233917LE, and M1206074763RE, as input observations. For depth supervision, we use the corresponding NAC DTM for each region. At each geometry step, we evaluate a cycled subset of the geometric consistency pairs rather than all pairs at once. For consistency among the original reference views, one ordered reference-to-reference direction is selected from the full set of ordered reference-view pairs. With three reference views, this cycles through all six directed reference-view pairs over six geometry steps. For consistency involving the orbit-augmented view, one original reference view is paired with the current orbit-augmented view at each geometry step. With three reference views, this cycles through all original reference views over three geometry steps.

Following the positioning of LNEM, we do not treat NAC DTM and SLDEM as direct baselines to be surpassed. Instead, we report them as production-level DEM references to show where the proposed neural reconstruction stands relative to existing lunar DEM products. The primary controlled comparison in this work is between our LNEM variant with geometric consistency and an ablated variant without the geometric consistency term. To isolate the effect of the proposed constraint, both variants use the same input observations, depth-supervision source, depth-weight schedule, and training configuration; the ablated model is obtained by setting the geometric consistency loss to zero while keeping all other settings unchanged.

## 4.2. Quantitative Results

For quantitative evaluation, we report $\text{RMSE}_{\text{corr}}$ with respect to valid LOLA reference points, following the LNEM bias-corrected evaluation protocol. We further evaluate geometric consistency using standard deviation maps computed from multi-view 3D point estimates.

Table 1 summarizes the quantitative evaluation results. We report bias-corrected RMSE with respect to valid LOLA reference points. The primary comparison in this table is between the model with the geometric consistency term and its ablated counterpart without the geometric consistency term, while NAC DTM and SLDEM are included as reference DEM products for contextual comparison.

The model with geometric consistency improves over the ablated baseline without geometric consistency, yielding lower $\text{RMSE}_{\text{corr}}$ in both evaluated regions. Because the two variants share the same depth-weight schedule and identical training configuration, this comparison directly isolates the effect of the proposed geometric consistency

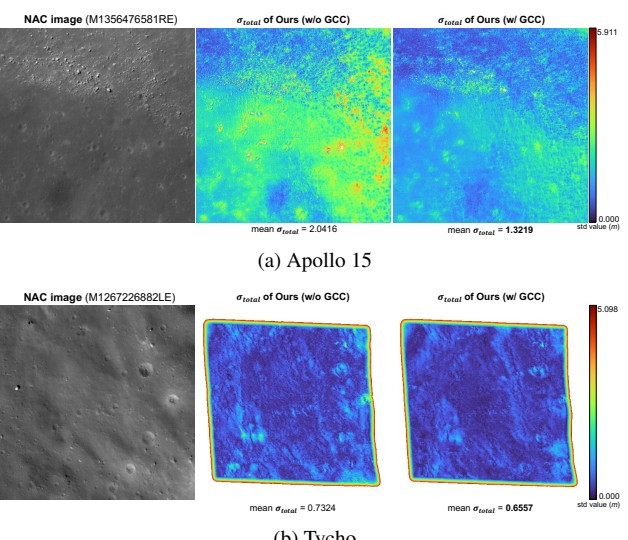

(a) Apollo 15

(b) Tycho

Figure 4. **Comparison of standard deviation maps with and without the geometric consistency constraint (GCC) across two lunar regions.** Each subfigure shows one lunar region. In each row, the left panel shows the reference NAC image, and the middle and right panels show $\sigma_{\text{total}}$ maps without and with GCC, respectively. The maps are computed from multi-view 3D point estimates, where $\sigma_{\text{total}} = \sqrt{\sigma_x^2 + \sigma_y^2 + \sigma_z^2}$. The model with GCC yields lower geometric variance, indicating more consistent terrain reconstruction across views.

term. These results indicate that the proposed constraint improves agreement with LOLA reference points within our framework.

Although NAC DTM and SLDEM provide useful production-level DEM references, our method is intended as a neural reconstruction framework rather than a direct replacement for production DEM pipelines. We therefore use these products to contextualize the absolute accuracy level of existing DEMs, while focusing our quantitative claim on the controlled improvement obtained by introducing the geometric consistency term.

## 4.3. Qualitative Results

Fig. 3 presents qualitative comparisons with SLDEM and NAC DTM to visualize the spatial characteristics of the learned neural elevation field. The proposed reconstruction exhibits continuous local terrain variations around structures visible in the input NAC observation.

Fig. 4 compares the geometric consistency maps with and without the proposed geometric consistency constraint. The maps are computed only over valid regions that are commonly observed by the reference views. In the Tycho case, the white boundary regions correspond to areas without sufficient common overlap among the three views and are therefore excluded from the variance computation.

| Reference image | Sampled views | Time |

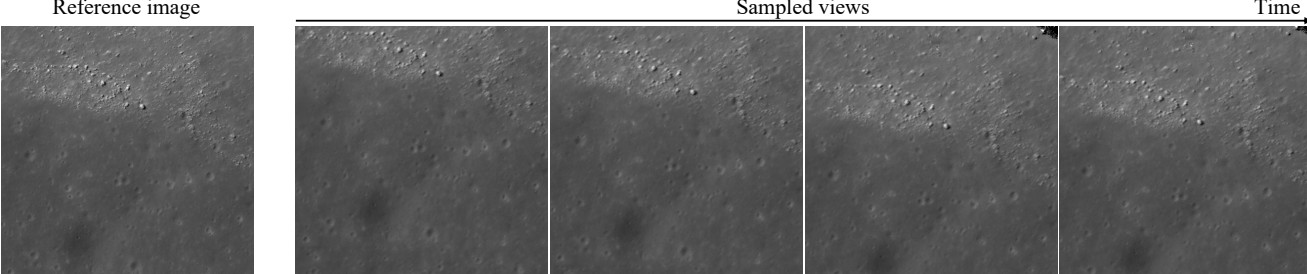

Figure 5. **Rendered sway frames from nearby viewpoints for NAC image M1356476581RE.** The leftmost image is the reference view from M1356476581RE, and the remaining images show rendered views along a nearby sway trajectory. Major terrain structures remain stable across adjacent rendered views, suggesting that the learned representation preserves coherent geometry under gradual viewpoint changes.

Within the valid overlap regions, the version with the proposed constraint produces lower variance across views, indicating that the geometric consistency constraint improves the agreement of reconstructed 3D geometry between orbital passes.

We further assess the spatial coherence of the learned representation through rendered sway frames from nearby novel viewpoints, as illustrated in Fig. 5. The rendered sequence exhibits smooth appearance transitions across the three training orbits and yields visually coherent results at unseen neighboring viewpoints. Major terrain structures remain stable under gradual viewpoint changes, suggesting that the learned representation preserves spatial continuity across adjacent pushbroom observations. This qualitative observation is consistent with the improved cross-view geometric consistency reported in the quantitative evaluation.

### 4.4. Discussion

The comparison between the model with geometric consistency and the ablated variant without it can be interpreted as a direct evaluation of the proposed geometric consistency term. Since both models use the same input observations, depth-supervision source, depth-weight schedule, ray sampling strategy, and training configuration within each region, the observed improvement is attributable to the geometric consistency constraint rather than to differences in the optimization setup.

Despite these improvements, several limitations remain. First, abrupt elevation artifacts can still appear near shadow boundaries and locally complex structures such as boulders, indicating that the reconstruction remains sensitive to strong illumination variation. Second, regions without sufficient common visibility across the reference views are excluded from the standard-deviation computation, as shown by the white boundary regions in the Tycho result. Third, the geometric consistency objective increases training-time overhead because additional rendering, reprojection, and interpolation operations must be performed across direc-

tional view pairs. Overall, the results support the usefulness of explicit cross-view geometric constraints for sparse-view pushbroom lunar reconstruction, while also highlighting the need for more robust treatment of illumination effects, overlap-aware evaluation, and more efficient geometric supervision.

### 5. Conclusion

In this paper, we addressed sparse-view lunar DEM reconstruction from pushbroom orbital imagery. To improve reconstruction stability under limited viewpoint diversity, we proposed a Gaussian orbit augmentation strategy together with a geometric consistency constraint. Our method combines Gaussian orbit augmentation, scanline-wise attitude repointing, continuous pushbroom reprojection, and a geometric consistency term to explicitly enforce cross-view consistency under line-wise imaging geometry.

Experiments on two LROC NAC regions, Apollo 15 and Tycho, showed that the proposed method improves geometric consistency and reduces bias-corrected reconstruction error with respect to LOLA reference points. Qualitative results further indicate that the proposed approach improves the spatial coherence of the learned elevation field and maintains stable terrain structures across nearby rendered viewpoints under sparse-view conditions. These results suggest that explicit cross-view geometric constraints are beneficial for lunar neural reconstruction from a limited number of pushbroom observations.

Despite these improvements, several challenges remain. The reconstruction can still be sensitive to strong illumination changes, particularly near shadow boundaries and abrupt local structures, and the additional geometric consistency objective increases computational cost during training. Future work will extend the method to more lunar regions and observation settings, and will investigate more robust handling of illumination effects and more efficient formulations of multi-view geometric supervision.

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
