# OpenReview forum: "Orbit-Augmented Geometric Consistency for Lunar Neural Elevation Model"
_thecvf.com/CVPR/2026/Workshop/3D4S — CVPR 2026 Workshop 3D4S Poster_

### Official Review · Reviewer_WGCr · 2026-04-16
**Incremental geometric consistency for pushbroom lunar reconstruction with limited validation**

**Rating:** 5
**Confidence:** 3

**Review:**

This paper presents an extension of the Lunar Neural Elevation Model (LNEM) for sparse-view lunar digital elevation model (DEM) reconstruction from pushbroom orbital imagery. The proposed method introduces orbit-aware data augmentation and a geometric consistency constraint based on bidirectional depth reprojection, aiming to improve reconstruction stability under limited viewpoint diversity.

The work is motivated by the challenges of sparse-view reconstruction in pushbroom imaging, where limited parallax and varying illumination conditions make cross-view geometric consistency difficult to enforce. The proposed approach incorporates Gaussian orbit augmentation and scanline-level modeling to preserve pushbroom geometry while introducing additional supervision through multi-view consistency constraints.

While the problem setting is relevant and the method is technically reasonable, the overall contribution appears limited. The proposed components are largely adaptations of existing ideas from neural rendering and multi-view geometry (e.g., data augmentation and reprojection-based consistency) to a specialized domain. As a result, the level of methodological novelty is relatively modest.

In addition, the experimental validation is somewhat limited, relying on a small number of input images and a single geographic region, which raises concerns about the robustness and generalizability of the approach. The quantitative improvements over the baseline are also relatively modest, further limiting the strength of the empirical evidence.

Overall, while the paper presents a coherent and technically sound extension in a domain-specific setting, the level of novelty and experimental validation may not be sufficient to meet the acceptance bar. I find the paper close to the threshold but slightly below it.

---

### Official Review · Reviewer_j4UL · 2026-04-24
**Solid extension of LNEM for pushbroom sensors, but empirical comparisons to classical baselines require nuance.**

**Rating:** 6
**Confidence:** 4

**Review:**

This paper addresses the challenge of high-resolution lunar digital elevation model (DEM) reconstruction from sparse orbital pushbroom imagery. The authors propose an extension to the Lunar Neural Elevation Model (LNEM) by introducing a Gaussian orbit augmentation strategy and a geometric consistency constraint. By applying rigid transformations to orbital trajectories and utilizing scanline-level attitude repointing, the method synthesizes novel views while preserving the underlying pushbroom imaging geometry. Furthermore, a bidirectional depth reprojection constraint is formulated to enforce cross-view consistency between reference and augmented observations.
Evaluation
Quality: The methodology is mathematically sound and rigorously tailored to the specific constraints of planetary orbital imaging. The formulation of the translational augmentation $\Delta t^{(k)}=s_{scene}\alpha(k)(b_{tr}+\epsilon_{tr})$ and the corresponding roll offset $\Delta\phi^{(k)}=\alpha(k)\epsilon_{\phi}$ accurately models orbital perturbations. The formulation of the geometric consistency loss across novel-to-real, real-to-real, and real-to-novel branches comprehensively regularizes the sparse-view optimization.
Clarity: The paper is well-structured and clearly written. The authors do an excellent job of distinguishing the line-wise acquisition geometry of linear pushbroom cameras from standard perspective pinhole camera models, effectively justifying why off-the-shelf sparse NeRF regularizations cannot be naively applied here.
Originality: Adapting novel-view synthesis regularizations to the specific dynamics of multiperspective pushbroom sensors is a valuable contribution. Instead of unconstrained rigid transformations, the approach of using progress-scaled base shifts tied to the orbit's forward axis demonstrates a deep understanding of the physical sensor constraints.
Significance: Reconstructing terrain from sparse LROC NAC data is a well-known bottleneck in planetary photogrammetry. While physically grounded, the actual empirical improvements over classical methods are currently limited, which tempers the immediate practical significance of the work.

Pros: The method explicitly and correctly models line-wise pushbroom geometry instead of relying on standard global projection centers or RPC-based approximations.The scanline-wise attitude repointing effectively anchors the virtual camera's focus on the original ground location, preventing geometric drift during augmentation.The ablation studies successfully isolate the proposed Geometric Consistency Constraint (GCC), demonstrating a measurable reduction in RMSE against LOLA altimetry (from 3.043m down to 2.906m).

Cons: The proposed method significantly underperforms compared to traditional baseline DEMs. The reported RMSE against LOLA is 2.906m, whereas classical NAC DTM achieves 1.918m and SLDEM achieves 2.115m. While the authors state this work is intended as a neural baseline rather than a production replacement, the performance gap raises questions about the practical utility of the neural approach in its current state.The experimental validation is limited in scope, focusing exclusively on a single location (the Apollo 15 landing site) using only three LROC NAC input images.The authors acknowledge that the geometric reprojection objective increases training-time computational overhead, but they fail to quantify this cost in the experimental results, making it difficult to assess the algorithm's efficiency.

---

### Decision · Program_Chairs · 2026-04-28

Accept (Poster)